# Predictors of dream enactment behavior among medical students: The case of the University of Gondar, Ethiopia

**Baye Dagnew**[1]*, **Mengistie Diress**[1], **Mihret Getnet**[1], **Mohammed Abdu Seid**[2], **Sofonias Addis Fekadu**[3], **Yibeltal Yismaw Gela**[1], **Yigizie Yeshaw**[1], **Yitayeh Belsti**[1], **Yonas Akalu**[1]

**1** Department of Human Physiology, School of Medicine, College of Medicine and Health Sciences, University of Gondar, Gondar, Ethiopia, **2** Unit of Human Physiology, Department of Biomedical Sciences, College of Health Science, Debre Tabor University, Debre Tabor, Ethiopia, **3** Department of Optometry, School of Medicine, College of Medicine and Health Sciences, University of Gondar, Gondar, Ethiopia

* bayedagnew7@gmail.com

## Abstract

### Introduction

Dream enactment behavior is one of the features of rapid eye movement sleep behavior disorder. It might be a manifestation of neurodegenerative diseases and can lead to fall associated injuries. There is no evidence of dream enactment behavior and its associated factors in Ethiopia. Hence, this study targeted to pinpoint the predictors of dream enactment behavior among Medical students at the University of Gondar.

### Methods

The cross-sectional survey was carried out at the University of Gondar among Medical students selected by simple random sampling technique from Dec 2020 to Feb 2021. We used a structured pretested questionnaire to collect the data and dream enactment behavior was evaluated using rapid eye movement sleep behavior disorder single question. Descriptive statistics were computed, and determinant factors were identified using binary logistic regression model. In the final model, explanatory variables with a p<0.05 were considered as predictors (statistically significant) of the dream enactment behavior. The strength of association was determined using adjusted odds ratio (AOR) with its 95% CI.

### Results

Four-hundred and twelve students took part in the study with 97.4% response rate. The mean age of participants was 20.82(±1.88) years and 291(70.63%) were males. The prevalence of dream enactment was 34.47% (95% CI: 30.02–39.20). Daytime sleepiness score (AOR = 1.104; 95% CI: 1.053–1.160), age (AOR = 1.15; 95% CI: 1.019–1.290), monthly pocket money (AOR = 0.9991; 95% CI: 0.9985–0.9997), alcohol drink (AOR = 2.71; 95% CI: 1.076–6.846), and perceived stress (AOR = 3.854; 95% CI: 1.802–8.242) were statistically significant factors of dream enactment behavior.

**Data Availability Statement:** All relevant data are within the paper and its Supporting Information files.

**Funding:** The author(s) received no specific funding for this work.

**Competing interests:** The authors have declared that no competing interests exist.

**Abbreviations:** AOR, Adjusted odds ratio; CI, Confidence interval; DEB, Dream enactment behavior.

## Conclusions

In this study, the magnitude of dream enactment behavior was high which was significantly associated with daytime sleepiness score, age, monthly pocket money, alcohol drink, and perceived stress all of which are modifiable except age. The University of Gondar has to plan a strategy to avert the condition via the prevention of the determinant factors. Students need to reduce stress and avoid alcohol drink. We strongly urge forthcoming scholars to ascertain association of dream enactment and academic performance of university students.

## Introduction

Dreaming is a sleep related cognitive activity characterized by multisensory imagery, emotional arousal and apparent speech and motor activity [1]. Dream enactment behavior (DEB) is one of the manifestations of rapid eye movement (REM) sleep behavior disorders [2]. Dream enactor flails arms, leaps from the bed, or crawl [3]. REM sleep suppresses neurotransmitters such as serotonin, norepinephrine, and histamine which in turn contributes for dreaming [4].

Dream enactment behavior might lead to fall associated injury [5, 6], and is also an early manifestation of neurodegenerative diseases [7]. A nearly 70% of enacted dreams partake threats to the victims [8]. Nocturnal injury is common in REM sleep behavior disorders because of the acting out of dreams [9]. Contrary to this, there are studies illustrating DEB is not a bad phenomenon [10]. Dream enactment behavior could be due to partly by striatal changes in the dopaminergic pathways [11]. The knowledge of DEB is helpful to design intervention and prevention strategies for neurological disorders [12]. Moreover, it has imminent function to prevent α-synucleinopathy-related motor and cognitive decline [13] and to innovate therapeutic options for clinicians [14]. Evidences showed violent behavior during sleep exist in 1.6% among adults of whom 78.7% reported vivid dreams and 31.4% hurt themselves or someone else [15]. Behavioral management has promising effects for dream disorders [9].

Dream enactment behavior is associated with other sleep disorders (such as sleepwalking and sleep terror), family history of the condition [15], socioeconomic status [16], and distress [17]. Inconsistent findings were reported about the association of DEB and sex [18–20]. Elderly people are more affected by DEB [21]. People with narcolepsy, severe sleep deprivation, hormonal changes, age older than 50 years, drug users, alcohol drinkers, and individuals with neurodegenerative diseases are at higher risk of DEB [22].

In Ethiopia, there are studies on sleep quality [23], excessive daytime sleepiness [24], REM sleep behavior disorders [25], and insomnia symptoms [26] among University students. However, there is no previous study in Ethiopia yet on this topic. Hence, we envisioned to determine the prevalence of DEB and examine the determinant factors among Medical students at the University of Gondar. This study will be helpful to plan precautionary strategies by identifying associated factors so that students will be competent and academically productive.

## Methods

### Study area, design, and period

Institution-based cross-sectional survey was carried out from Dec 2020 to Feb 2021 at the University of Gondar, College of Medicine and Health Sciences, Northwest Ethiopia. University of

Gondar is among the pioneer universities in Ethiopia serving the country since 1954. According to the statements written at the University website (www.uog.edu.et), currently the College of Medicine and Health Science has over 7,000 students and offers 22 undergraduate and 38 postgraduate programs.

## Sample size and sampling technique

Sample size was determined using single population proportion formula with the following assumptions: proportion of dream enactment behavior (p = 0.5, no previous study), 95% confidence interval (CI), and 5% margin of error (d). The minimum sample size was 384 and after adding 5% non-response rate, we found a total of 423 samples. We applied simple random sampling technique to recruit study participants. The samples were from each year of study (from 1st year to final year) using proportional allocation by year of study. Over 1400 Medical students were enrolled in the same academic year of the data collection period.

## Population

The source population was all undergraduate Medical students at the College of Medicine and Health Sciences. As described above, the facilitators enter to each class (in each year of students from year 1 to year 5). In each classroom the facilitators explained the purpose of the study. Then students were recruited by computer generated simple random sampling technique using excel spreadsheet. All medical students who came to the class during the data collection period were used as study population. We excluded students with severe illness (to avoid exhaustion) from the survey.

## Data collection

Self-administered structured questionnaire was used to collect the required data for this study. The questionnaire comprised sociodemographic variables (sex, age in years, year of study, pocket money in Ethiopian birr), bedtime routines (Facebook use, coffee consumption, bathing, physical exercise, and alcohol drink), daytime sleepiness score, perceived stress (dichotomized), and the outcome variable (DEB). Two Postgraduate Psychiatry students served as facilitators after oriented about ethical issues and purpose of the study by the investigators.

## Measurement of the study variables

We used a Single-Question Screen for REM sleep Behavior Disorder (RBD1Q) [27] which has only a single question, with two responses "Yes" or "No". The participants were asked "Have you ever been told, or suspected yourself, that you appear to 'act out your dreams' while asleep such as flailing your arms in the air, punching, and running movements)? This tool is framed as better screening tool for epidemiological studies. Perceived stress scale (PSS-10) which was validated in Ethiopia for the assessment of stress [28] was used to measure stress. We used Epworth sleepiness scale to determine daytime sleepiness.

## Data quality control

Careful processes were performed starting from the design of the questionnaire. Because the questionnaire was not validated in Ethiopia, we invited epidemiologists to assure the validity of the questionnaire and we finally used it after incorporating the comments from the experts. Pretest was done among 35 students at Maraki, one of the campuses of the University of Gondar. Finally, the facilitators strictly checked the filled questionnaire for completeness and consistency.

## Statistical analysis

We entered the data using Epidemiological information version 7 and then exported into Stata 16 for data cleaning, recoding, and analysis. Normal distribution was checked using skewness and normality tests. For description of continuous variables, mean with standard deviation and range, as well as median and interquartile range (IQR) were computed. Frequency with percent was executed to describe categorical variables. Chi-square and t-tests were used to compare categorical and continuous variables, respectively, between students with DEB and non-DEB. Bivariable (at $p<0.2$) and multivariable (at $p<0.05$) binary logistic regression were performed to identify determinant factors of dream enactment. The degree of association was described using the adjusted odds ratio with 95% CI.

## Ethical approval and consent to participate

Ethical approval was obtained from the Ethical committee of the School of Medicine, University of Gondar. Each participant gave written informed consent, and no identifiers were recorded.

# Results

## Descriptive characteristics of students by dream enactment behavior

A total of 412 students took part in the study with a response rate of 97.4%. Two hundred and ninety-one (70.63%) students were males, 160 (38.83) were first-year students, and 340 (82.52%) were orthodox Christian. We checked normal distribution for continuous variables, and we found that age, perceived stress score and Epworth sleepiness score were normally distributed whereas monthly pocket money was not normally distributed (right skewed). The mean age of participants was 20.82 (range: 18–34) years. The mean values of daytime sleepiness score and perceived stress score were 8.69 (±4.70) and 18.58(4.97), respectively. The median value of pocket money was 500 ETB (IQR = 2900).

Dream enactment behavior was higher in frequency among firs-year students (36.25%), males (36.08%). As observed from Table 1, 339(82.28%) participants reported the presence of perceived stress of which 133 (39.23) reported DEB. Twenty-six (6.31%) students drunk alcohol of which 16 (61.54%) had DEB. The chi-square and T-test results showed the association of DEB with age, perceived stress, daytime sleepiness score, and pocket money (Table 1).

The prevalence of dream enactment behavior among the participants was 34.47% (95% CI; 30.02–39.20) (Fig 1).

## Predictors of dream enactment behavior among medical students

We conducted bivariable binary logistic regression at $p<0.2$ for all explanatory variables. From these, daytime sleepiness score, age of the respondent, monthly pocket money, coffee consumption and alcohol drink immediately before sleep, and perceived stress were associated with dream enactment behavior without adjustment to other variables. Then, we executed multivariable binary logistic regression analysis to adjust for other variables to eliminate the association due to chance at $p<0.05$. Of the candidate variables in the bivariable analysis, daytime sleepiness score, age, monthly pocket money, alcohol drink, and stress were statistically significant factors of dream enactment behavior (Table 2).

A unit increases in daytime sleepiness score increases the odds of DEB by 10.4% (AOR = 1.104; 95% CI: 1.052–1.160). The odd of DEB is elevated by 14.7% (AOR = 1.146; 95% CI: 1.019–1.290) for a unit increase of age in years. Monthly pocket money is inversely associated with the occurrence of DEB in that a unit increase in monthly pocket money reduces

**Table 1. Descriptive characteristics by dream enactment behavior of Medical students of University of Gondar, Northwest Ethiopia, 2021.**

| Variables | | Study sample (N = 412) | Dream enactment behavior | | $X^2$/t-test p-value |
|---|---|---|---|---|---|
| | | | No (%) | Yes (%) | |
| Sex [f (%)] | Male | 291 (70.63) | 186 (63.92) | 105 (36.08) | 0.284 |
| | Female | 121 (29.37) | 84 (69.42) | 37 (30.58) | |
| Study year [f (%)] | First-year | 160 (38.83) | 102 (63.75) | 58 (36.25) | 0.544 |
| | ≥Second | 252 (61.17) | 168 (66.67) | 84 (33.33) | |
| Religion [f (%)] | Orthodox | 340 (82.52) | 216(63.53) | 124 (36.47) | 0.342 |
| | Protestant | 37 (8.98) | 32 (86.49) | 5 (13.51) | |
| | Muslim | 26 (6.31) | 17 (65.38) | 9 (34.62) | |
| | Others | 9 (2.18) | 5 (55.56) | 4 (44.44) | |
| Perceived stress [f (%)] | No (scored <15) | 73 (17.72) | 64 (87.67) | 9 (12.33) | 0.000 |
| | Yes (scored ≥15) | 339 (82.28) | 206 (60.77) | 133 (39.23) | |
| Facebook use [f (%)] | No | 158 (38.35) | 102 (64.56) | 56 (35.44) | 0.742 |
| | Yes | 254 (61.65) | 168 (66.14) | 86 (33.86) | |
| Alcohol drinks [f (%)] | No | 386 (93.69) | 260 (67.36) | 126 (32.64) | 0.003 |
| | Yes | 26 (6.31) | 10 (38.46) | 16 (61.54) | |
| Age in years [c], mean (SD), range | | 20.82 (1.88), 18–34 | 20.58 (1.70) | 21.27 (2.12) | 0.0004 |
| Pocket money (ETB) [c], mean (SD), range | | 684.7(507.6), 100–3000 | 750.6 (570.8) | 559.5(324.6) | 0.0003 |
| Epworth sleepiness score [c], mean (SD), range | | 8.69 (4.70), 0–24 | 7.80 (4.17) | 10.34 (5.19) | 0.000 |
| Perceived stress score [c,] mean (SD), range | | 18.58 (4.97), 1–38 | 17.47 (4.76) | 20.69 (4.69) | 0.000 |

f = Frequency, c = continuous variable, ETB = Ethiopian birr, SD = Standard deviation.

DEB by 0.09% (AOR = 0.9991; 95% CI: 0.9985–0.9997). The odd of DEB is 2.71 times higher (AOR = 2.71; 95% CI: 1.076–6.846) among students who drink alcohol immediately before sleep compared to their counterparts. Students who reported perceived stress had 3.85 times (AOR = 3.854; 95% CI: 1.802–8.242) fold DEB compared to the references.

## Discussion

Most people experience DEB at sleep time particularly during the REM sleep stage. Academic stressors could impose students to be stressed and have sleep disturbances. We examined the

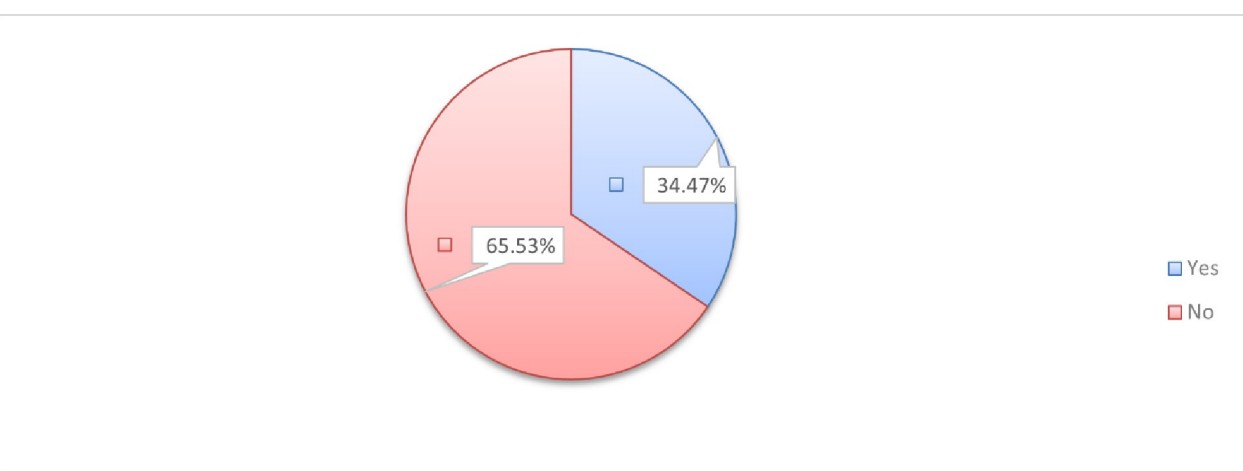

**Fig 1. Magnitude of DEB among medical students at the University of Gondar, Northwest Ethiopia, 2021.**

**Table 2. The determinants of DEB using binary logistic regression model among medical students at the University of Gondar, Northwest Ethiopia, 2021 (N = 412).**

| Variables | Unadjusted | Adjusted |
|---|---|---|
| | OR (95% CI) | OR (95% CI) |
| ESS[c] | 1.124(1.074–1.177) | **1.104(1.053–1.160)** *** |
| Age in years[c] | 1.218(1.086–1.367) | **1.146(1.019–1.290)** * |
| Pocket money (ETB)[c] | 0.9990(0.9984–0.9995) | **0.9991(0.9985–0.9997)** ** |
| Coffee drink before sleep (Yes) | 1.538(0.826–2.866) | 0.860 (0.412–1.795) |
| Alcohol drink before sleep (Yes) | 3.302(1.457–7.483) | **2.71462(1.076–6.846)** * |
| Perceived stress (Yes) | 4.591(2.210–9.537) | **3.854323(1.802–8.242)** *** |

ESS = Epworth sleepiness scale; c = Continuous; ETB = Ethiopian birr; OR = Odds ratio; bold fonts indicate statistically significant factors with dream enactment behavior

*significant at p<0.05

**significant at p<0.01

***significant at p<0.001.

determinant factors of DEB among Medical students at the University of Gondar. To the best of our knowledge, this study is the first in its kind in Ethiopia. In this study, the prevalence of DEB was 34.47% (30.02–39.20) which was significantly associated with daytime sleepiness score, age of the participants, monthly pocket money, alcohol drink, and perceived stress level. The existence of DEB among university students may be caused partly due to academic associated anxiety and seep disruption [29]. This finding is higher than another study in Japan migraine patients [30] but lower than a study among university students in Canada (66%) [18]. The difference might be due to sample size, and the population of interest. The prevalence of DEB in our study is lower than studies among other population as observed in post-partum women (63%) and pregnant (40%) women [22] whereas our finding is far higher than studies in the general adult population (5.9% in men, and 4.1% in women), older adults (10.9%) [31]. This difference might be accounted for the varying activities of the population. In Ethiopia, there was a study among University students to determine REM sleep behavior disorder which was 46.25% [25]. This is higher than our finding. This might be because the current study was conducted among Medical students and the previous one was among health and medical students, besides, the tool was different.

The odd of DEB is elevated by 14.7% for a year increase in age. There is one study with similar finding [32]. As age increases, there will be increased daily stressful situations which might lead to dreaming in memorizing the daily activities and thoughts since dreaming is believed to be partly by the theory of biological response, organization of knowledge, activation-synthesis, and threat-simulation [33]. REM sleep behavior disorder is higher in older people which can increase the existence of DEB [34]. Monthly pocket money is inversely associated with the occurrence of DEB. Reports revealed that low socioeconomic status mediates the existence of stressors and hence higher income reduces DEB because students with high income may fulfill their economic needs that potentially prevent stress, anxiety, and depression [35].

A unit increase in daytime sleepiness score increases the odds of dream enactment behavior. This is supported by previous studies [30, 36]. This association might be because people chronically sleep deprived present higher scores for ESS, in turn, shorter latency for REM sleep, which increases the chance for DEB [37].

Though the numbers of people who drink alcohol were very small i.e 26 students, we found significant association between alcohol drink and DEB. The odd of DEB is 2.71 times among students who drink alcohol immediately before sleep compared to their counterparts which is

in line with another study [38]. This might be because of the effect of alcohol on subsequent sleep disruption and fragmented sleep [39]. Besides, alcohol withdrawal associated-orexin gene reduction may lead to daytime sleepiness and hence DEB will occur during sleep at night [40]. Alcohol drink initially increases non-REM sleep by increasing gamma-amino butyric acid (GABA) and, later on, as the blood alcohol level drops off, REM sleep period increases and hence DEB elevates [41]. Students who reported perceived stress had 3.85 times folds DEB compared to the references. This is in line with other studies [42–44]. The association of stress with DEB is explained by the changes in cortisol hormone produced by the adrenal cortex. As a coping mechanism, cortisol is produced during stressful situations. In return, cortisol elevation plays a role to increase the content and nature of dreaming [45].

Even though few scholars argue the role of dreaming in memory consolidation [46], it is embarrassing and exerts bad health impact to the victims. This study establishes the need to institute screening of DEB among students and identifying the potential risk factors. Longitudinal studies are required to establish strong evidence on the associated factors of DEB and interventions has to be designed based on the findings. As a limitation, the tool used to determine the outcome variable is a screening instrument which cannot be used as a diagnostic modality. The cross-sectional survey did not infer cause-effect relationship.

## Conclusions

In this study, the magnitude of DEB was high which was associated with modifiable factors except age. The University of Gondar needs to plan strategies to avert the condition via the prevention of the determinant factors such as launching protocols to restrict alcohol drink and reduce stress. Students must avoid alcohol drink immediately before sleep and reduce their stress level using different modalities. We strongly urge the forthcoming scholars to ascertain the association of DEB and academic performance of university students.

## Supporting information

**S1 Data.**
(DTA)

## Acknowledgments

We acknowledge the University of Gondar and study participants.

## Author Contributions

**Conceptualization:** Baye Dagnew, Mengistie Diress, Mihret Getnet, Mohammed Abdu Seid, Sofonias Addis Fekadu, Yibeltal Yismaw Gela, Yigizie Yeshaw, Yitayeh Belsti, Yonas Akalu.

**Data curation:** Baye Dagnew, Sofonias Addis Fekadu, Yibeltal Yismaw Gela, Yigizie Yeshaw, Yonas Akalu.

**Formal analysis:** Baye Dagnew, Mengistie Diress, Mihret Getnet, Mohammed Abdu Seid, Yigizie Yeshaw, Yonas Akalu.

**Investigation:** Baye Dagnew.

**Methodology:** Baye Dagnew, Mengistie Diress, Mihret Getnet, Mohammed Abdu Seid, Sofonias Addis Fekadu, Yigizie Yeshaw, Yitayeh Belsti, Yonas Akalu.

**Resources:** Baye Dagnew.

**Software:** Baye Dagnew, Mengistie Diress, Mohammed Abdu Seid, Yigizie Yeshaw.

**Supervision:** Baye Dagnew.

**Validation:** Baye Dagnew.

**Visualization:** Baye Dagnew, Mengistie Diress, Mihret Getnet, Sofonias Addis Fekadu, Yigizie Yeshaw, Yonas Akalu.

**Writing – original draft:** Baye Dagnew, Mengistie Diress, Mihret Getnet, Mohammed Abdu Seid, Sofonias Addis Fekadu, Yibeltal Yismaw Gela, Yigizie Yeshaw, Yitayeh Belsti, Yonas Akalu.

**Writing – review & editing:** Baye Dagnew, Mengistie Diress, Mihret Getnet, Mohammed Abdu Seid, Sofonias Addis Fekadu, Yibeltal Yismaw Gela, Yigizie Yeshaw, Yitayeh Belsti, Yonas Akalu.

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
