## [Decision Letter · Decision Letter 0]

18 Nov 2021

PONE-D-21-16361Predictors of dream enactment behavior among Medical students: The case of the University of Gondar, EthiopiaPLOS ONE

Dear Dr. Dagnew,

Thank you for submitting your manuscript to PLOS ONE. After careful consideration, we feel that it has merit but does not fully meet PLOS ONE’s publication criteria as it currently stands. Therefore, we invite you to submit a revised version of the manuscript that addresses the points raised during the review process.

We look forward to receiving your revised manuscript.

Kind regards,

Sidarta Ribeiro

Academic Editor

PLOS ONE

Journal Requirements:

3. Please amend the manuscript submission data (via Edit Submission) to include author Yigizie Yeshaw.

Reviewers' comments:

Reviewer's Responses to Questions

**Comments to the Author**

1. Is the manuscript technically sound, and do the data support the conclusions?

Reviewer #1: Yes

Reviewer #2: Yes

2. Has the statistical analysis been performed appropriately and rigorously? 

Reviewer #1: No

Reviewer #2: Yes

3. Have the authors made all data underlying the findings in their manuscript fully available?

Reviewer #1: Yes

Reviewer #2: Yes

4. Is the manuscript presented in an intelligible fashion and written in standard English?

Reviewer #1: Yes

Reviewer #2: Yes

5. Review Comments to the Author

Reviewer #1: PONE-D-21-16361

Title: Predictors of dream enactment behavior among Medical students: The case of the University of Gondar, Ethiopia

This manuscription is research about a cross-sectional survey that was carried out at the University of Gondar, Ethiopia, with medical students who was selected by simple random sampling technique. To evaluated dream enactment the authors used rapid eye movement sleep behavior disorder single question. They used a binary logistic regression model that after adjusted odds ratio, they found dream enactment behavior was high which was significantly associated with daytime sleepiness score, age, monthly pocket money, alcohol drink, and perceived stress all of which are modifiable except age.

We agree with the statistical test that author used, but I have some consideration.

1. Didn´t show results of normal distribution test.

2. Why did Bivariable (at p<0.2?) and multivariable (at p<0.05) binary logistic regression were use?

3. In case of perceived stress the data is not balance (17 versus 83 %). The same for alcohol drinks. It could be explained in the text.

4. Did the medical students at University of Gondar have sleep medicine class? It could influence the behavior of the student and it could be one of strategies to avert this condition.

The research results need to be better described, especially in relation to the distribution and unbalance in the number of cases for some variables. This could be one of the factors that may be masking the results. I suggest that this issue be addressed by the authors so that the manuscript can be in a format for publication.

Reviewer #2: Review of paper: Predictors of dream enactment behavior among Medical students: The case of the University of Gondar, Ethiopia

PlosOne

Overview: this is a very interesting study aiming to evaluate the prevalence of dream enacting behavior (DEB) on the Medical students population from an university of Ethiopia. Furthermore, authors also investigated the predictors of DEB by means of binary logistic regression models and calculated the OR for significant predictors. Their findings indicate a moderate prevalence (34%) of DEB on the population. Similar prevalence has been described elsewhere. Interesting, they also found associations between stress (the higher, the worse), money pocket (the less, the worse), and sleepiness (the higher, the worse) with the prevalence o DEB. In my opinion this is a well conducted experiment, very well written with interesting results. The most important thing I wish authors to discuss a bit more is why they chose to use only a single question for DEB instead of a classic questionnaire (REM sleep behavior disorder screening questionnaire, for example) for a better screening of DEB? I am questioning this because authors decided to include the Epworth Sleepiness Scale (which has no validation for their country).

Major

Methods

Sampling: authors described that a “simple random sampling technique” was applied. I understand it, however, I would like to know better about the recruitment process. How were participants recruited to take part into the survey? At line 90 page 4 there is a statement “Students who were present at the time of data collection were included in the study”. I apologize, but I could not understand what authors mean with “student who were present”. Present where? This is related to the recruitment process already mentioned.

Data Quality control: since author mentioned a quality control procedure, I am wonder how the consistency was determined by the facilitators?

Dream enactment behavior assessment: author assessed DEB by means of a single question. Also, authors claimed that this is a “betters screening tool for epidemiological studies”. Could authors provide a comparison between their question and other tools for assessing REM-sleep behavior disorders, or just DEB?

Results:

It was a bit surprising to see that over 70% of the sample is male. Could authors provide an explanation for it? Is it related to the willing to take part into the study, or is it representative of the university students’ population?

Discussion:

page 8 lines172, 173: I do not agree with the assumption that REM sleep increases in magnitude as a function of age. It is well known that REM sleep is reduced as people get old. Please, investigate Li et al 2018. Sleep in Normal Aging. Sleep Med. Clin.

Page 9 lines 178-180: I think authors went a bit too far to explain the association between sleepiness and their findings on DEB. Perhaps a simpler explanation would be that people chronically sleep deprived present higher scores for ESS, in turn, shorter latency for REM sleep, which increases the chance for DEB.

Also, I missed a discussion about the prevalence of DEB on their population in comparison to others. I would suggest authors to explore a bit more the review (already cited) from Baltzan et al 2020. There is research on REM-sleep behavior disorder that must be discussed in this manuscript.

Finally, I would be a bit more conservative on interpreting their findings from alcohol consumption since only 26 participants declared to drink. Authors should acknowledge this when discuss this finding.

Minor

Introduction:

At page 3 line 62 “However, there is another study where sex is insignificant”. I think authors mean that sex is not significantly associated. I would not say “insignificant”. Furthermore, there is a typo, perhaps authors mean “are” instead of “is”.

Methods:

Ethics: It is very clear the ethical statement at the checklist, but I would advise authors to include similar statement on ethical approval and written informed consent by participants at the methods section.

6. PLOS authors have the option to publish the peer review history of their article (what does this mean?). If published, this will include your full peer review and any attached files.

Reviewer #1: **Yes: **John Fontenele Araujo

Reviewer #2: **Yes: **Felipe Beijamini

---

## [Author Response · Author response to Decision Letter 0]

16 Dec 2021

Point-by-point responses to the academic editor and reviewers 

MS ID: PONE-D-21-16361

Title: Predictors of dream enactment behavior among Medical students: The case of the University of Gondar, Ethiopia

We are very grateful for giving us the chance for the revision and improvement of our manuscript. All the authors involved in the revision process and we hope we addressed all the raised issues. The changes we made are marked red in the revised manuscript.

Academic editor’s comments 

Authors’ reply: Thank you. We assured that we fulfill the PLOS ONE’s requirements. 

2. In your Data Availability statement, you have not specified where the minimal data set underlying the results described in your manuscript can be found. 

Authors’ reply: Thank you. We included a statement describing the minimal dataset in the revised manuscript. We also uploaded the study’s minimal data set. This is described in the cover letter.

3. Please amend the manuscript submission data (via Edit Submission) to include author Yigizie Yeshaw.

Authors’ reply: We amended as suggested. 

Authors’ reply: We relocated the Ethics statement from the declaration to the Methods section as recommended. 

Authors’ reply to Reviewer #1: PONE-D-21-16361

This manuscription is research about a cross-sectional survey that was carried out at the University of Gondar, Ethiopia, with medical students who was selected by simple random sampling technique. To evaluated dream enactment the authors used rapid eye movement sleep behavior disorder single question. They used a binary logistic regression model that after adjusted odds ratio, they found dream enactment behavior was high which was significantly associated with daytime sleepiness score, age, monthly pocket money, alcohol drink, and perceived stress all of which are modifiable except age.

We agree with the statistical test that author used, but I have some consideration.

1. Didn´t show results of normal distribution test (line 134-137) 

Authors’ reply: Thank you very much. Sorry for not including in the results section. We actually conducted normal distribution test before the start of the overall analysis. Monthly pocket money was not normally distributed (right skewed) whereas age, perceived stress score and Epworth sleepiness score were normally distributed. We included this in the revised manuscript. 

2. Why did Bivariable (at p<0.2?) and multivariable (at p<0.05) binary logistic regression were use? 

Authors’ reply: Thank you. In the first phase of the binary logistic regression (bivariable analysis) we checked the crude association between the outcome variable (DEB) and each single independent variable. To avoid possible elimination of important variables (statistically), we used a p<0.2 as cut-off value. Those variables with a p<0.2 were candidates and entered into the multivariable logistic regression analysis. Finally, as a common recommendation of statisticians we used a p<0.05 as cut of point (in the multivariable regression) to decide statistical significance of each variable by keeping other variables constant. 

3. In case of perceived stress the data is not balance (17 versus 83%). The same for alcohol drinks. It could be explained in the text.

Authors’ reply: Thank you. We calculated the percent (%) in table 1 as; for the first column (total), we calculated percentages for column (column %) e.g. for stress from the total of 412 students 133 (39.23%) had stress. For the next columns, the percent was calculated as row %. As observed here, 339(82.28%) participants reported the presence of perceived stress of which 133 (39.23) reported dream enactment behavior. The same works for alcohol drink and other variables. Twenty-six (6.31%) students drunk alcohol of which 16 (61.54%) had DEB.

4. Did the medical students at University of Gondar have sleep medicine class? It could influence the behavior of the student and it could be one of strategies to avert this condition.

Authors’ reply: Thank you. As you describe sleep medicine class affects the behavior of students but there is no such course in the curriculum of medicine for undergraduate program in this university. However, students when they are in the first year of their study, they may took sleep physiology which we believe all students took it. Therefore, we expect equal course delivery regarding sleep physiology and all the included departments attended physiology course because it is given in the first semester of the first year of their study period.

The research results need to be better described, especially in relation to the distribution and unbalance in the number of cases for some variables. This could be one of the factors that may be masking the results. I suggest that this issue be addressed by the authors so that the manuscript can be in a format for publication.

Authors’ reply: Thank you. We explored all the texts for the mismatch with the values in the table and we made corrections accordingly in the revised manuscript. 

Authors’ reply to Reviewer #2: 

Overview: this is a very interesting study aiming to evaluate the prevalence of dream enacting behavior (DEB) on the Medical students population from an university of Ethiopia. Furthermore, authors also investigated the predictors of DEB by means of binary logistic regression models and calculated the OR for significant predictors. Their findings indicate a moderate prevalence (34%) of DEB on the population. Similar prevalence has been described elsewhere. Interesting, they also found associations between stress (the higher, the worse), money pocket (the less, the worse), and sleepiness (the higher, the worse) with the prevalence of DEB. In my opinion this is a well conducted experiment, very well written with interesting results. The most important thing I wish authors to discuss a bit more is why they chose to use only a single question for DEB instead of a classic questionnaire (REM sleep behavior disorder screening questionnaire, for example) for a better screening of DEB? I am questioning this because authors decided to include the Epworth Sleepiness Scale (which has no validation for their country).

Authors’ reply: Thank you very much for the kind appreciation of our topic and the way we describe. We tried to address all the comments raised as follows. 

Reviewer’s concern: Major

Methods

Sampling: authors described that a “simple random sampling technique” was applied. I understand it, however, I would like to know better about the recruitment process. How were participants recruited to take part into the survey? At line 90 page 4 there is a statement “Students who were present at the time of data collection were included in the study”. I apologize, but I could not understand what authors mean with “student who were present”. Present where? This is related to the recruitment process already mentioned.

Authors’ reply: Thank you very much for raising this issue. We already did it but we didn’t included in the previous version of our manuscript. It gives sense if included in the manuscript which persuaded as to agree with the comment and hence we included the following in the revised manuscript. “As described above, the facilitators enter to each class (in each year of students from year 1 to year 5). In each class room the facilitators explained the purpose of the study. Then students were recruited by computer generated simple random sampling technique using excel spreadsheet”. Sorry for creating confusion by “those who were present”, by this we mean that “All medical students who came to the class during the data collection period were used as study population”. All this are detailed in the revised manuscript. 

Data Quality control: since author mentioned a quality control procedure, I am wonder how the consistency was determined by the facilitators?

Authors’ reply: Thank you. The facilitators checked completeness and consistency. By consistency, we mean that facilitators checked the records of students if there were illogical records. E.g. a student may fill his age as 119 mistakenly to write 19 and so on. Another thing is students who responded “No” to “I have or had the following phenomena during my dreams” may list the activities which we expect to be empty. Therefore, facilitators were responsible to check such issues. 

Dream enactment behavior assessment: author assessed DEB by means of a single question. Also, authors claimed that this is a “betters screening tool for epidemiological studies”. Could authors provide a comparison between their question and other tools for assessing REM-sleep behavior disorders, or just DEB?

Authors’ reply: REM Sleep Behavior Disorder Single-Question Screen (RBD1Q), a screening question for dream enactment with a simple yes/no response with a sensitivity of 93.8% and a specificity of 87.2% (as reported by Ronald B Postuma et al 2012) which are comparable to other REM screening tools with longer questionnaires. That is why this instrument is considered as better epidemiological tool.

Results: It was a bit surprising to see that over 70% of the sample is male. Could authors provide an explanation for it? Is it related to the willing to take part into the study, or is it representative of the university students’ population?

Authors’ reply: The high number of males in our study is due to the large proportion of male medical students in the university. This could be due to in Ethiopian context, joining medical schools usually requires being academically top scorer and due to other social burdens beyond academic duties (helping their family etc) female students may not get a chance to join medical schools. That is why we have less number of female participants. 

Discussion:

page 8 lines172, 173: I do not agree with the assumption that REM sleep increases in magnitude as a function of age. It is well known that REM sleep is reduced as people get old. Please, investigate Li et al 2018. Sleep in Normal Aging. Sleep Med. Clin.

Authors’ reply: Thank you very much. We looked into the given reference and other related articles as well. We agree that REM sleep time (normal REM period) is reduced as people get old but the NREM sleep increases as the person gets older especially in the adult population. When we look at the abnormal REM i.e. REM sleep behavior disorder, it increases as people get older. Older people have higher rate of REM sleep behavior disorder. In the revised manuscript we described as REM sleep disorder. We then revised the discussion accordingly. 

Page 9 lines 178-180: I think authors went a bit too far to explain the association between sleepiness and their findings on DEB. Perhaps a simpler explanation would be that people chronically sleep deprived present higher scores for ESS, in turn, shorter latency for REM sleep, which increases the chance for DEB.

Authors’ reply: Thank you very much for the compliment. We included the statement as suggested. 

Also, I missed a discussion about the prevalence of DEB on their population in comparison to others. I would suggest authors to explore a bit more the review (already cited) from Baltzan et al 2020. There is research on REM-sleep behavior disorder that must be discussed in this manuscript.

Authors’ reply: Thank you very much. We included the recommended statements in the revised manuscript. 

Finally, I would be a bit more conservative on interpreting their findings from alcohol consumption since only 26 participants declared to drink. Authors should acknowledge this when discuss this finding.

Authors’ reply: Thank you very much. We included the following statement in the discussion section before we discussed the odds of alcohol drink to give clue about the small observations of alcohol drink to the audiences “Though the numbers of people who drink alcohol were very small i.e 26 students, we found significant association between alcohol drink and DEB.”. 

Minor

Introduction:

At page 3 line 62 “However, there is another study where sex is insignificant”. I think authors mean that sex is not significantly associated. I would not say “insignificant”. Furthermore, there is a typo, perhaps authors mean “are” instead of “is”.

Authors’ reply: Thank you. Regarding sex, we merged it with previous statement as “there were inconsistent findings about the association between DEB and se then we listed the references. We check all the typos errors and we hope we improved the revised manuscript. 

Methods:

Ethics: It is very clear the ethical statement at the checklist, but I would advise authors to include similar statement on ethical approval and written informed consent by participants at the methods section.

Authors’ reply: Thank you very much. The editor advised us to relocate the ethics statement into the methods section. And hence, we relocated the ethics stamen into methods section.

---

## [Decision Letter · Decision Letter 1]

31 Jan 2022

Predictors of dream enactment behavior among Medical students: The case of the University of Gondar, Ethiopia

PONE-D-21-16361R1

Dear Dr. Dagnew,

We’re pleased to inform you that your manuscript has been judged scientifically suitable for publication and will be formally accepted for publication once it meets all outstanding technical requirements.

Kind regards,

Sidarta Ribeiro

Academic Editor

PLOS ONE

Additional Editor Comments (optional):

Reviewers' comments:

Reviewer's Responses to Questions

**Comments to the Author**

1. If the authors have adequately addressed your comments raised in a previous round of review and you feel that this manuscript is now acceptable for publication, you may indicate that here to bypass the “Comments to the Author” section, enter your conflict of interest statement in the “Confidential to Editor” section, and submit your "Accept" recommendation.

Reviewer #1: All comments have been addressed

Reviewer #2: All comments have been addressed

2. Is the manuscript technically sound, and do the data support the conclusions?

Reviewer #1: Yes

Reviewer #2: Yes

3. Has the statistical analysis been performed appropriately and rigorously? 

Reviewer #1: Yes

Reviewer #2: Yes

4. Have the authors made all data underlying the findings in their manuscript fully available?

Reviewer #1: Yes

Reviewer #2: Yes

5. Is the manuscript presented in an intelligible fashion and written in standard English?

Reviewer #1: Yes

Reviewer #2: Yes

6. Review Comments to the Author

Reviewer #1: The authors answer all questions and the manuscript is in publication format. Then I agree that the manuscript is accepted

Reviewer #2: Thank you, authors, for answered to all my previous queries. The manuscript is significantly improved.

There is only a typo at line 140 page 6. It is written “firs-year”.

7. PLOS authors have the option to publish the peer review history of their article (what does this mean?). If published, this will include your full peer review and any attached files.

Reviewer #1: **Yes: **John F Araujo

Reviewer #2: **Yes: **Felipe Beijamini

---

## [Editor Report · Acceptance letter]

15 Feb 2022

PONE-D-21-16361R1 

Predictors of dream enactment behavior among Medical students: The case of the University of Gondar, Ethiopia 

Dear Dr. Dagnew:

I'm pleased to inform you that your manuscript has been deemed suitable for publication in PLOS ONE. Congratulations! Your manuscript is now with our production department. 

Kind regards, 

on behalf of

Sidarta Ribeiro 

Academic Editor

PLOS ONE